# Spiking neural state machine for gait frequency entrainment in a flexible modular robot

**Alex Spaeth**[1,2]*, **Maryam Tebyani**[1], **David Haussler**[2,3], **Mircea Teodorescu**[1,2]

**1** Department of Electrical and Computer Engineering, University of California, Santa Cruz, Santa Cruz, California, United States of America, **2** Genomics Institute, University of California, Santa Cruz, Santa Cruz, California, United States of America, **3** Howard Hughes Medical Institute, University of California, Santa Cruz, Santa Cruz, California, United States of America

* atspaeth@ucsc.edu

**Data Availability Statement:** All relevant data are within the manuscript and its Supporting Information files.

**Funding:** This research was supported by a grant made to the Braingeneers research group by

## Abstract

We propose a modular architecture for neuromorphic closed-loop control based on bistable relaxation oscillator modules consisting of three spiking neurons each. Like its biological prototypes, this basic component is robust to parameter variation but can be modulated by external inputs. By combining these modules, we can construct a neural state machine capable of generating the cyclic or repetitive behaviors necessary for legged locomotion. A concrete case study for the approach is provided by a modular robot constructed from flexible plastic volumetric pixels, in which we produce a forward crawling gait entrained to the natural frequency of the robot by a minimal system of twelve neurons organized into four modules.

## 1 Introduction

Artificial intelligence has in recent years seen a revolution in the area of "neuromorphic computation", the use of brain-inspired computational units which, like biological neurons, communicate with each other using discrete events called "spikes" to perform a variety of tasks [1]. These techniques can have significant advantages in terms of power consumption and the ability to make use of the temporal structure of input data [2].

It is common to derive spiking neural networks directly from deep learning approaches, either by modifying the learning technique to be applicable to a spiking neural network, or by globally converting a traditional neural net into an equivalent spiking neural network [3]. Both of these approaches have shown promise on many problems, including the generation of complex behaviors for simple robots [4–6]. In this paper, we aim to illustrate a very different approach, where spiking neurons are used as elements to design a circuit which implements a desired behavior.

The function of interest in our application is a central pattern generator, or CPG, inspired by the way in which the simplest organisms such as nematodes use neurons to implement periodic motor patterns and to make simple decisions such as retreating in response to a toxic

Schmidt Family Futures. The funders had no role in study design, data collection and analysis, decision to publish, or preparation of the manuscript.

**Competing interests:** This research was supported by a grant made to the Braingeneers research group by Schmidt Family Futures. This does not alter our adherence to PLOS ONE policies on sharing data and materials.

chemical stimulus [7]. Pattern generation is a well-studied topic in computational neuroscience [8], but in robotics, central pattern generators constructed from spiking neurons are relatively rare. Instead, applications typically use the abstract concept of a CPG, but construct periodic motions from periodic primitives such as phase oscillators to produce motion commands [9]. However, an approach informed by neuroscience is worthwhile because robotic experiments in this vein can generate interesting insights about the design tradeoffs made by real neural systems [10].

Applications of this principle to robotics include a variety of open-loop spiking CPGs generating position commands for robots ranging from hexapods [11] to fish [12]. Also, certain researchers have decided to tackle the problem of proprioceptive feedback in quadrupeds [13–15]. In this paper, which extends work previously presented at the 2020 IEEE International Conference on Soft Robotics [16] using a prototype of the same robot, we will demonstrate the design and implementation of a spiking central pattern generator entrained by proprioceptive feedback to control the locomotion of a modular robot. This system serves as a case study for a simple conceptual framework which allows us to implement closed-loop robotic control through a minimal spiking neural network without recourse to black-box optimization.

The organization of this paper is as follows. Section 2 describes the basic module making up the controller as well as how such modules can be used to implement arbitrary state machines. Section 3 demonstrates that the computational properties of this module are insensitive to parameter variation yet relatively easy to modulate with the appropriate inputs. Section 4 applies these principles to generate a specific desired state machine for the control of a flexible robot. Finally, Section 5 describes the experiments that were carried out on this platform.

## 2 Building blocks for universal computation

From a design perspective, it can be helpful to treat neurons as asynchronous digital logic elements which represent an ON state by firing a train of action potentials at a given rate, and OFF by remaining quiescent. This simplified model necessarily ignores many subtleties of the underlying analog neural system, and it is not Turing complete [17], but it can be used to implement neural state machines for robotic control [18].

In this section, we propose a basic module which can be used to construct such a state machine in the setting of neuromorphic computation: a three-neuron circuit which can hold state, depicted in Fig 1. This module is a neuromorphic implementation of an electronic set-reset (SR) latch, a component which holds one bit of state controlled by its two inputs, "set" (S) and "reset" (R). Like its electronic prototype, this circuit can switch between two states depending on its input: an inactive resting state representing a saved logical value of 0, and an active spiking limit cycle representing a logical 1. We can construct such a circuit by connecting two neurons, labeled $E_1$ and $E_2$ in the figure, in strong mutual excitation; in this configuration, whenever either neuron fires, the other will fire in response, but the system can remain in a quiescent state for any duration if neither neuron is externally induced to fire.

Technically the described behavior is not tonic firing, although the time series of spikes appears that way. Tonic firing occurs when a neuron fires repeatedly in response to a constant current or due to its own intrinsic dynamics, whereas the neurons in this module only produce single action potentials in response to presynaptic firings. In that sense, this module is an oscillator constructed from two neurons each individually incapable of oscillation. Similarly, when the module state is toggled by outside stimuli, its behavior may appear superficially similar to bursting, but the underlying dynamics are fundamentally different [19]. This mechanism of rhythmogenesis is certainly not typical for biological central pattern generation [11], but it is effective from an engineering perspective.

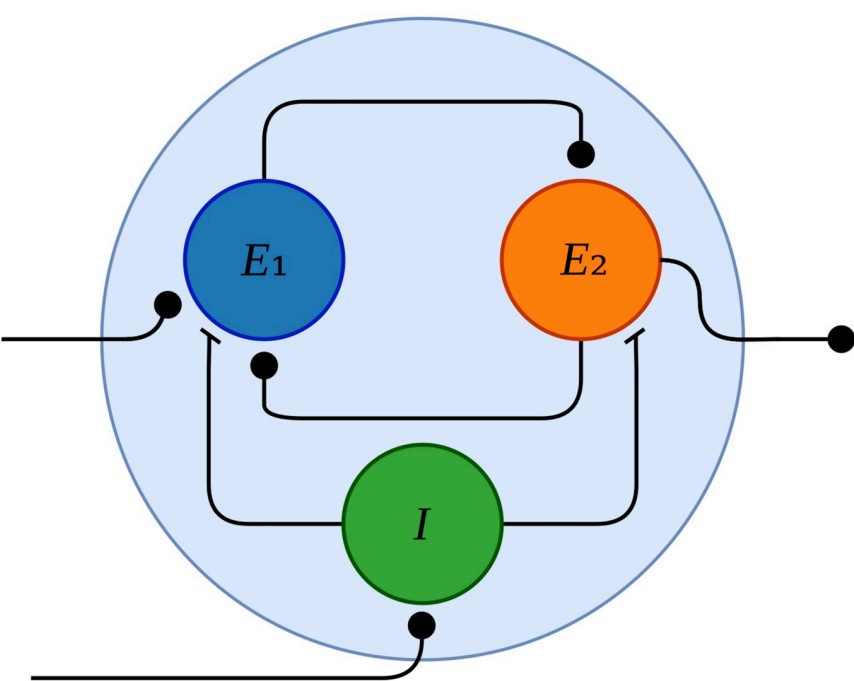

**Fig 1. The neural latch module.** Arrowheads are circular (flat) to represent excitatory (resp. inhibitory) connections. The symmetrically connected pair of excitatory neurons $E_1$ and $E_2$ are connected to produce positive feedback. An excitatory input to $E_1$ on the left activates the module, and the output is taken from $E_2$ on the right. The inhibitory interneuron $I$ can deactivate the module in response to an external signal which enters on the bottom left.

Biological neurons are classified broadly as either excitatory or inhibitory, and inhibitory interactions are typically short range, so to complete the circuit we introduce a local inhibitory interneuron $I$ within the module, which we call the "reset interneuron". This neuron, which provides the only inhibitory inputs to $E_1$ and $E_2$, can receive a long-range excitatory connection to function as the "reset" input in the SR latch metaphor. Weak excitatory synapses from $E_1$ and $E_2$ to $I$ allow the latch to modulate its own reset: an external input just barely strong enough to trigger a reset while the latch is active will not trigger a reset while the latch is inactive, conserving the energy of the interneuron.

The activity of the three neurons in the neural module is depicted in Fig 2. The excitatory neurons $E_1$ and $E_2$ fire in alternation until an externally-induced firing of the reset interneuron $I$ silences them. Note that each excitatory neuron fires only when the other provides enough synaptic current to activate it. Activity stops because the final pulse of synaptic input from $E_2$ to $E_1$ is curtailed by the single firing of $I$. The effect of this inhibition is not obvious in the plot of synaptic currents due to the slow time constant of inhibition in this model; it is visible in the shape of the tail more so than in the height of the peak. However, even this small change is sufficient to stop $E_1$ from firing, deactivating the module.

## 2.1 Breaking the digital mold

Although it can help in the conceptual design phase to view neural systems through a digital lens, such models can be overly limiting in terms of efficiency and expressiveness. Many functions which would require a very large number of logic gates to implement using binary neurons can be realized in a relatively straightforward way using the analog properties of a much smaller number of spiking neurons.

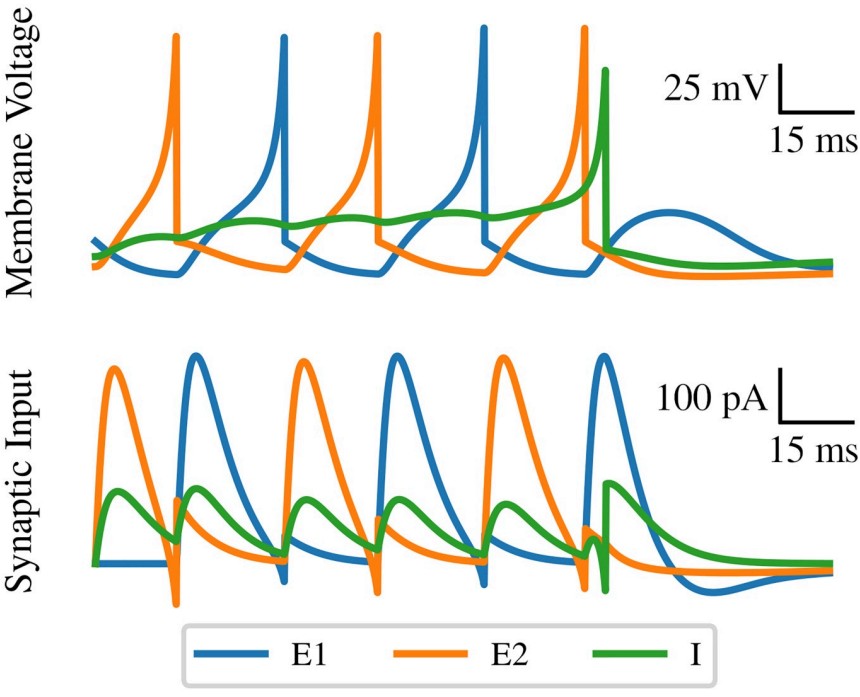

**Fig 2. Activity in the spiking neural module.** Membrane voltage and input synaptic current for each neuron are displayed as the module is deactivated by some external stimulus. The three neurons are represented by their membrane voltages, with three traces corresponding to the color code established in Fig 1: the two excitatory neurons are shown in blue and orange, while the inhibitory neuron is in green.

From an electrophysiological perspective, a neuron is an electrically active cell with a variety of ionic channels which maintain the membrane voltage near a resting value but can generate large spike-shaped excursions called "action potentials" in response to currents induced by input neurotransmitters. Spiking neural models are those which reproduce this excitability in the form of an excitable dynamical system.

A good intuitive model of this behavior is given by the leaky integrate-and-fire (LIF) neuron, in which the membrane voltage $v$ has linear dynamics tending towards a fixed resting potential $V_r$, but if the collective effect of the inputs drives the membrane voltage above the peak value $V_p$, a spiking event is said to have occurred, and the membrane voltage is reset to a fixed value $c$.

As an example of how these dynamics may be useful, the simplest realization of a delay in a digital circuit is a series of many identical short delay elements such as inverters, whereas an analog system can implement a delay through parameter tuning without introducing new circuit elements. A small excitatory input can gradually increase the membrane voltage of a neuron until it fires, so that a sequence of many presynaptic firings at sufficiently high frequency is required before any signal is propagated to the output. In fact, it is exactly this method which we will use in Section 2.3 to ensure that the activation of each neural module propagates only gradually to other modules to which it is connected.

Instead of viewing the neural module within the digital logic metaphor, we can take the dynamical perspective and treat it as a bistable relaxation oscillator. If a neuron does not fire, it does not contribute to the positive feedback loop responsible for the oscillatory state, so the state of the module as a whole remains unchanged for subthreshold inputs. On the other hand,

for inputs large enough to produce a spike, there is positive closed-loop gain, producing oscillatory behavior.

It is worth noting that in the context of continuous dynamics, the analogy to a state machine is made by a so-called "heteroclinic network", where the active limit cycle of a module represents an attractor corresponding to a state [20]. Inputs, disturbances, or weak coupling between modules can move the system state from one attractor to another, producing a state transition along a heteroclinic trajectory.

Other dynamical properties of the neuron may also be used for computation. The morphological diversity of the real neuron seems to be used to great advantage in this context: dendritic computation and the availability of a wide array of different neurotransmitters and classes of inhibition can allow the implementation of logic gates at a finer granularity than the individual cell [7]. Likewise, it is possible for the dynamics of individual neurons to produce bistability between resting and tonic firing or between tonic firing and bursting [21]. Additionally, besides the "integrator-type" neurons which we have described conceptually, neurons also exist which are electrically resonant, responding preferentially to inputs which come at a certain frequency [19]. However, we will not be exploring these phenomena in this paper.

## 2.2 Modeling spiking neurons

In order to take advantage of the analog computational properties of spiking neurons, we apply the Izhikevich model together with conductance-based synapses to describe the behavior of our simulated neurons. We have chosen to adapt Izhikevich's spiking neural model because it trades off phenomenological accuracy with efficient computation in a reasonable way [22]. This model is essentially a LIF neuron augmented with two main features: a quadratic nonlinearity to reproduce the dynamics of the biological threshold behavior, and the recovery variable $u$, which abstracts various slow ionic currents to provide persistent state across spikes. The model parameters are physically interpretable, and the model dynamics can be fit to observed electrophysiological data.

Although the Izhikevich model is not the simplest neural model capable of producing a bistable oscillator network [23], its accurate recapitulation of electrophysiological activity is important from a design perspective. Qualitatively similar behavior can be generated by a variety of different systems, but greater fidelity to biological prototypes helps a system designer consider constraints comparable to those faced in the evolution of biological organisms.

Once individual neurons have been modeled, the next consideration is how to connect them in networks. Large neural networks commonly employ a Dirac delta synapse, where each firing of a presynaptic neuron instantaneously increases the membrane voltage of the postsynaptic neuron. However, phase synchronization of integrator-type neurons requires synaptic activity with finite nonzero duration [24]. For this reason, we implement conductance-based synapses, where each synaptic connection is represented as a conducting channel gated by a presynaptic activation $x$, which increases after a firing event and gradually decays back to zero. When activated, the channel admits a current proportional to the presynaptic activation times the difference between the postsynaptic membrane potential and the synaptic reversal potential [25].

The time course of the presynaptic activation follows the classic "alpha" function $x(t) = \frac{t}{\tau} e^{-t/\tau}$ observed for synaptic conductances in vivo [26]. This function is the solution to the second-order linear differential equation $\tau^2 \frac{d^2 x}{dt^2} + 2\tau \frac{dx}{dt} + x = 0$, with the initial conditions $x(0) = 0$ and $\tau \frac{dx}{dt}\big|_{t=0} = 1$, so separating into two first-order differential equations by introducing a second variable $y = \tau \frac{dx}{dt}$, we obtain $x$ and $y$ dynamics with which the Izhikevich model can be augmented to produce the following dynamical model of a single neuron subject to an input

current $I(t)$:

$$
\begin{aligned}
\text{When } v < v_p \qquad C\tfrac{\mathrm{d}}{\mathrm{d}t}v &= k(v - v_r)(v - v_t) - u + I(t) \\
\tfrac{\mathrm{d}}{\mathrm{d}t}u &= a(b(v - v_r) - u) \\
\tfrac{\mathrm{d}}{\mathrm{d}t}x &= j/\tau \\
\tfrac{\mathrm{d}}{\mathrm{d}t}y &= -(2y + x)/\tau
\end{aligned}
\tag{1}
$$

$$
\begin{aligned}
\text{When } v = v_p \qquad v &\leftarrow c \\
u &\leftarrow u + d \\
y &\leftarrow y + 1
\end{aligned}
$$

After equipping these neurons with the parameters described in Table 1, we can combine them into neural networks consisting of $N$ neurons coupled to each other only through the input current $I(t)$. The magnitude of this current is parameterized by a matrix $G$ whose entries $g_i^j$ specify the peak input synaptic conductance triggered in the $i$th neuron by a firing in the $j$th neuron. The synaptic reversal potential $V_n$ is also specified per presynaptic neuron. The resulting input $I_i(t)$ to the $i$th neuron is therefore given by:

$$
I_i(t) = \sum_{j=1}^{N} g_i^j x_j (V_{n,j} - v_i)
\tag{2}
$$

Finally, we generate an actuation effort command for the physical actuators using simple simulated muscle cells. Because our system is designed with invertebrate control schemes in mind, it is reasonable to use a muscle model where, unlike the electrically active vertebrate muscle cell, muscle cells do not fire action potentials. Instead, they simply provide contractile force in proportion to their depolarization [27]. The result is similar to applying an exponential low-pass filter to the spiking waveform [11], smoothing the output of the system so that changes occur only on the timescale of a few milliseconds, determined by the synaptic and membrane time constants.

**Table 1. Parameters for the Izhikevich neuron.**

| Parameter | | Cell Type | | Units |
|:---:|:---|:---:|:---:|:---:|
| | | RS | LTS | |
| $a$ | characteristic rate of $u$ | 0.03 | 0.03 | ms$^{-1}$ |
| $b$ | leakage conductance | -2 | 8 | nS |
| $c$ | downstroke return voltage | -50 | -53 | mV |
| $d$ | downstroke inrush current | 100 | 20 | pA |
| $C$ | membrane capacitance | 100 | 100 | pF |
| $k$ | sodium channel gain | 0.7 | 1.0 | nS mV$^{-1}$ |
| $V_r$ | resting potential | -60 | -56 | mV |
| $V_t$ | threshold voltage | -40 | -42 | mV |
| $V_n$ | synaptic reversal potential | 0 | -70 | mV |
| $V_p$ | action potential peak | 35 | 20 | mV |
| $\tau$ | synaptic time constant | 5 | 20 | ms |

## 2.3 Choosing model parameters

We have designed the neural latch of Fig 1 according to a simplified binary neural model, whereas the spiking dynamical neural model with which we are simulating our circuit requires all of the various parameters summarized in Table 1. The question then arises of how one should choose the values of these parameters in order to obtain some desired behavior.

   We use the parameter values shown in Table 1, corresponding to well-characterized human cell types from the literature [19]. Excitatory cells are modeled as regular spiking (RS) pyramidal cells, and inhibitory cells take parameters corresponding to low-threshold spiking (LTS) inhibitory interneurons. These parameters have been chosen mainly based on their ready availability and the fact that they do not have intrinsic bursting dynamics or bistability, allowing us to implement the essential features of the module in circuitry rather than depending on exotic dynamics. However, it is fully possible to use parameters corresponding to other cell types in order to implement the same behaviors. For example, if we had used a set of parameters which produces bursting dynamics, our module might have implemented a limit cycle where the two neurons alternate bursts rather than spikes, but the same logical operations and constructions would have been available to us. As we will see later, this structural style of design leads to robust behavior in the module, in contrast with the precise tuning that would typically be necessary if all of our logical operations were implemented using such analog properties.

   With these values given, only the following synaptic conductances must be tuned manually in order to implement any desired logical structure: the strong connection $G_{exc}$ between the two excitatory neurons within the module, another relatively strong synapse $G_{inh}$ allowing the reset interneuron $I$ to deactivate the module, a weak synapse $G_{rst}$ priming the reset to fire while the module is active, a relatively weak feedforward input $G_{ffw}$ by which one module might gradually activate another, and a moderately strong connection $G_{fb}$ by which one module can quickly deactivate another. We found empirically that these parameters can be set with only a small amount of trial-and-error fine tuning due to their wide tolerances, as we will see in Section 3.1. Furthermore, three of them are internal to the module, and so robustness in these parameters is inherited by larger networks constructed from the same building block. The values we arrived at are given in Table 2.

   The neuron parameters affect the behavior of the module, but to varying degree. For instance, the action potential peak $V_p$ can be set almost completely arbitrarily because by the peak of an action potential, $v$ is on a trajectory which would reach infinity in only a few milliseconds if not for the spike reset. As a result, the choice of precisely where to clip the spike does not substantially alter the trajectory [19].

   Other parameters have a non-negligible effect on the period of the oscillation within the module, but for many of these, since their main effect is essentially shifting the timescale on which the neuronal dynamics play out, they have much less effect on the quantities which we want to control, such as the activation thresholds and necessary synaptic conductances. For

**Table 2. Synaptic parameters for the neural latch module.**

| Parameter and Use | | Value (nS) |
|---|---|---|
| $G_{exc}$ | connection within the module | 20 |
| $G_{inh}$ | strength of the "reset" interneuron | 10 |
| $G_{rst}$ | priming the reset during module activity | 5 |
| $G_{ffw}$ | gradual activation of other modules | 10 |
| $G_{fb}$ | deactivation of other modules | 10 |

example, increasing the after-spike reset voltage $c$ essentially gives the dynamics of the neuron a head start along the trajectory that must be traversed between each firing, meaning that the period decreases, but because it does not affect the continuous dynamics, this has no effect at all on the process of initiating the active state.

More pertinent to activation behavior, the strength of the excitation from one neuron to another is controlled by several synaptic parameters: the synaptic time constant $\tau$, the peak synaptic conductance $G_{\mathrm{exc}}$, and the synaptic reversal potential $V_n$ all control the shape and scale of the postsynaptic potential initiated by a firing event. A synaptic activation with a longer duration, higher peak conductance, or higher synaptic reversal potential will produce a larger or longer postsynaptic potential, and therefore cause a postsynaptic firing either more quickly or more surely.

One may imagine the neuron as a simple circuit with a single capacitance representing the cell membrane, and transmembrane conductances connecting this capacitor to voltage sources corresponding to ionic reversal potentials. In particular, the synaptic conductance connects to the reversal potential $V_n$. This view is the basis of biophysical models of neuronal dynamics (see, for example, ref. [25]), but as a mental image it can potentially be misleading because it encourages visualizing the steady state of an RC circuit, where the actual value of the conductance is unimportant; in fact, the synaptic conductance is only nonzero for a short time relative to the RC time constant, generating a small deviation in the voltage $v$ across the capacitor. As a result, the value of the conductance actually has a substantial impact on the amplitude of the postsynaptic potential, directly affecting whether or not the membrane voltage surpasses the threshold. Beyond this minimum, there is a wide range of acceptable values for this conductance.

## 3 Robustness of the neural module

Biological central pattern generation is carried out by small networks of neurons which must be robust to certain types of parameter variation in order for their behavior to withstand changes in the environment, but at the same time must be sensitive to other changes so that their behavior can be modulated by other components of the nervous system [28]. Although certain organisms may utilize the redundancy of large cell populations in neural circuits to mitigate unavoidable variations in neuronal parameters [29], our constructed neural system is not redundant and so must be intrinsically robust.

In this section, we quantify the behavior of the neural latch module of Fig 1 with respect to variations in the parameters of the two excitatory neurons. First, dynamical analysis is applied to measure the maximum deviation in each parameter for which the existence of the spiking limit cycle is preserved. Next, we use a Monte Carlo approach to quantify the sensitivity of the module to simultaneous variation in all twelve parameters of both neurons.

### 3.1 Preservation of the spiking limit cycle

The essential function of the module is to switch between resting and spiking states in response to external input. In the absence of bounds on the strength of the input, as long as both states exist and continue to be attractive, it is always possible to switch between them. Therefore, we operationalize the question of whether a given parameter set produces a viable module by checking for the existence and stability of both attractors.

The existence of the resting state can be checked by finding a fixed point of the dynamics, where all nullclines in the system intersect; since three of the four phase variables of each neuron obey linear dynamics, a fixed point can exist only when both neurons have $(u, x, y) = (b(v - V_r), 0, 0)$. Under this condition, since the synaptic activation variable $x$ is zero, the neurons

are decoupled, and the quadratic dynamics of the single remaining phase variable $v$ have roots at $v = V_r$ and $v = V_t + {}^b/_k$. Varying the parameters $b$, $k$, $V_r$, and $V_t$ produces a transcritical bifurcation when ${}^b/_k = -(V_t - V_r)$. For these parameter values, the two fixed points coincide, resulting in a single nonhyperbolic half-stable fixed point. However, for all other values of the parameters, the continuous dynamics exhibit exactly one stable node separated from the firing threshold by the unstable manifold of a saddle point, meaning that an attractive resting state always exists, although it is only marginally stable at a single pathological point in the parameter space.

Since the resting state is always attractive, only the existence of the attractive limit cycle must be verified for different parameter sets. However, it is nontrivial to prove the existence of a limit cycle, especially for a discontinuous dynamical system such as this one. Instead, we determine acceptable parameter values by numerically approximating a discrete dynamical system called the Poincaré first return map. This map is defined from an underlying continuous dynamical system by choosing a manifold $\mathcal{M}$ in phase space known to be transversal to flow. The Poincaré map $P$: $\mathcal{M} \to \mathcal{M}$ is then defined by following the dynamics from each point on $\mathcal{M}$ to the next point where the flow intersects $\mathcal{M}$. Any fixed point of $P$ lies on a periodic orbit in the original system, which is an attracting limit cycle if the fixed point is attractive. This is explained in more detail in any standard text on dynamical systems theory (e.g. chapter 10 of ref. [30]).

Our Poincaré section $\mathcal{M}$ is the hyperplane $v_1 = v_p$. As a result, the Poincaré map $P$ takes the state vector at each firing of the first neuron to its value the next time the first neuron fires. We compute this map numerically in the attached Julia simulation code using the ODE solver library DifferentialEquations.jl [31], then find its fixed point by Picard iteration, the process of iteratively applying the map until the results converge, starting at an arbitrary point on the Poincaré section. Fixed points found in this manner are attractive, meaning that they correspond to stable limit cycles. We also checked that if the limit cycle exists, its period is more than 5 ms so that the dynamics will remain numerically stable when computed on the robot, where we must use fixed-timestep integration with $\Delta t$ in the hundreds of microseconds. However, this requirement did not affect the range of acceptable parameters.

We first check for the persistence of the spiking limit cycle when the module is subject to variations in individual parameters, with both excitatory neurons modulated identically. The computed acceptable parameter ranges are given in Table 3. Parameter differences between

**Table 3. Parameter ranges for the neural latch.**

| Param. | Min. | Nom. | Max. | Units |
|:---:|:---:|:---:|:---:|:---:|
| $a$ | 0.017 | 0.030 | $\infty$ | ms$^{-1}$ |
| $b$ | −6.08 | −2.0 | 0 | nS |
| $c$ | −∞ | −50.0 | −40.3 | mV |
| $d$ | 40.4 | 100 | 169 | pA |
| $C$ | 68.3 | 100 | 159 | pF |
| $k$ | 0 | 0.70 | 1.41 | nS mV$^{-1}$ |
| $V_r$ | −66.3 | −60.0 | −29.2 | mV |
| $V_t$ | −47.0 | −40.0 | −36.2 | mV |
| $V_p$ | −32.7 | 35.0 | $\infty$ | mV |
| $V_n$ | −9.7 | 0.0 | 9.7 | mV |
| $\tau$ | 3.77 | 5.0 | 7.41 | ms |
| $G_{\text{exc}}$ | 16.1 | 20.0 | 31.6 | nS |
| $G_{\text{rst}}$ | 0.0 | 5.0 | 7.1 | nS |
| $G_{\text{inh}}$ | 4.4 | 10.0 | $\infty$ | nS |

the two neurons typically lead to asymmetries in the spiking pattern, where the two neurons take different amounts of time to activate, although this effect does not interfere with the functionality of the module. Other undesirable effects include overexcitation causing desynchronization between the two neurons, leading to a chaotic spiking attractor rather than a limit cycle. In principle the module could still perform its function under these conditions, but it is so different from our intended behavior that we consider it defective. $G_{exc}$, $\tau$, and $V_n$ have their own interesting failure mode. As the firing rate increases, for a given value of $G_{rst}$, the neuron can trigger its own reset, breaking up the limit cycle. When $G_{rst}$ is set to zero, these parameters can be set as large as numerical stability will allow.

Our prior statement that it is always possible to switch between the two attractive states if they both exist is only true if outside input to the module is unconstrained. However, inhibition to a module should come only from its own reset interneuron. The result is one additional constraint: the conductance $G_{inh}$ must be strong enough to move the module from the spiking limit cycle to the resting state. It is simple to verify this using the Poincaré map techniques already developed. Specifically, rather than ensuring that under the given parameter values a spiking initial condition eventually converges to the limit cycle, we ensure that if $G_{rst}$ is increased enough that the module triggers its own reset, the resulting synaptic activity is sufficient to break the limit cycle.

When this is the case, we already know that any outside input sufficiently strong to cause either of the neurons $E_1$ or $E_2$ to fire will bring the module to its spiking limit cycle. Furthermore, we have established that an outside input sufficiently strong to trigger a firing of the reset interneuron will bring the module back to its resting state. This process therefore establishes both the bistability of the system and the feasibility of switching between the two attractors in response to external input which is constrained to be excitatory.

### 3.2 Monte carlo analysis

The method of the previous section identifies the maximum independently acceptable deviation in each individual parameter value, but concurrent deviations may interact in nontrivial ways. For example, a lower resting voltage $V_r$ or a higher threshold $V_t$ makes the resting state more attractive, but this can be compensated by increasing the synaptic reversal potential $V_n$.

In order to generalize to simultaneous variations in all neuron parameters, we use a Monte Carlo approach to study fractional parameter variation. We introduce a fractional variation totaling 10% by selecting points $\xi$ from a 11-dimensional standard Gaussian distribution and rescaling to set $\|\xi\| = 1$. We then multiply each of the parameters by 10% of the base value of the corresponding parameter, and add this value to the initial parameter vector. This method selects a random parameter variation totaling exactly 10%, but because $\xi$ has been sampled uniformly from the surface of the unit sphere, its components are not independent: a large deviation in one direction typically corresponds to smaller deviations in other parameters.

For illustrative purposes, a few examples of the typical effect of this type of parameter variation on the behavior of the module are demonstrated in Fig 3. The membrane voltage of the two excitatory neurons during the spiking limit cycle of the unmodified module is compared to three different modules, each with all parameters of both neurons $E_1$ and $E_2$ randomly adjusted by 10%. In each of these cases, although the relative timing of spiking events varies substantially, the main qualitative features are identical: the two excitatory neurons alternate firing in a consistent pattern.

Although Fig 3 demonstrates three cases where random parameter variation did not affect the qualitative behavior of the module, this is not always the case. Sometimes, the variant module is not viable in that its dynamics converge globally to the resting state. Whether a module

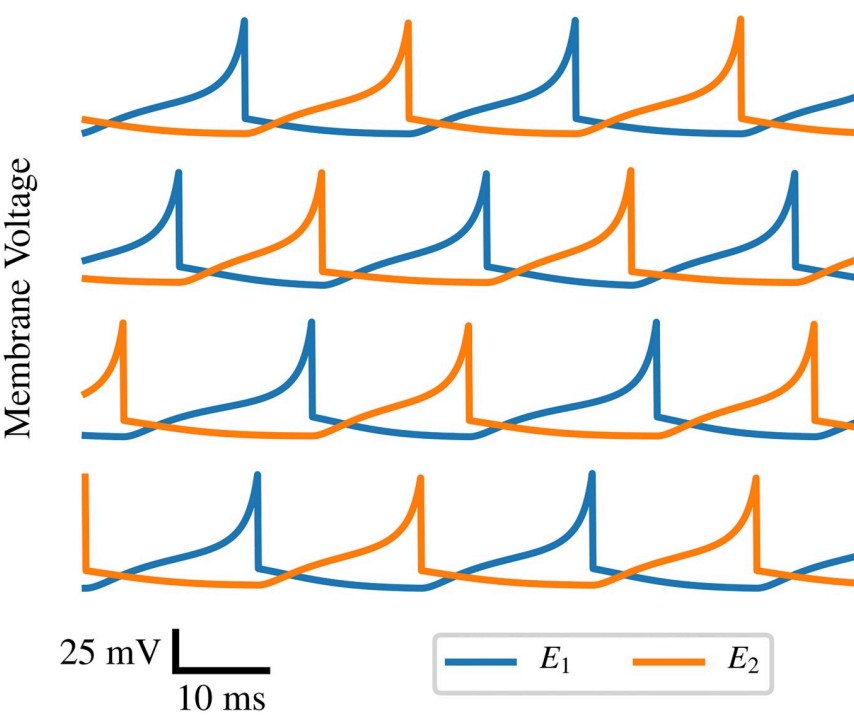

**Fig 3. Effect of random parameter variation.** The module was simulated four times with randomized parameter values. The four plots represent the membrane voltage of the two excitatory neurons $E_1$ and $E_2$ in these four independent experiments. The top trace corresponds to the nominal parameter values; the lower three are the result of random modulation of all parameters of both neurons. The traces are qualitatively identical, and differ only in details of spike timing.

functions correctly or not depends on the parameter change which occurs. We quantified these effects with a Monte Carlo approach, where one million random variant modules were tested to approximate the probability with which an introduced 10% parameter variation causes the module to no longer function. This test was run in three separate variants: modifying only one of the two neurons led to failure in 1.3% of cases, whereas applying the same modification to both neurons led to failure in 2.8% of cases, and modifying both neurons independently led to failure in 5.4% of cases.

Variations in different parameters have different importance. Many of the parameters have little effect on the observable dynamics of the neurons when only small deviations are considered, while other parameters may be close to a threshold beyond which the module cannot function. To study this effect, we consider a point cloud of the twelve-dimensional relative deviation variables $\xi$. These are unitless random variables, generated as described above, which specify the relative deviation in each parameter. The points in this cloud are labeled according to whether the module continued to function in the sense of Section 3.1 when subjected to that deviation. Since this point cloud is symmetric, dimensionality reductions which revolve around spatial clustering, such as principal component analysis (PCA), are not applicable. Instead, we use our labels for linear discriminant analysis (LDA), a simple machine learning method based on the optimization problem of finding the hyperplane which best separates two label classes. In this case, we are not trying to classify our points, but we can use the hyperplane for visualization. The normal vector of this hyperplane is a basis vector in the higher-dimensional space such that the projection of the point cloud onto this axis is maximally informative.

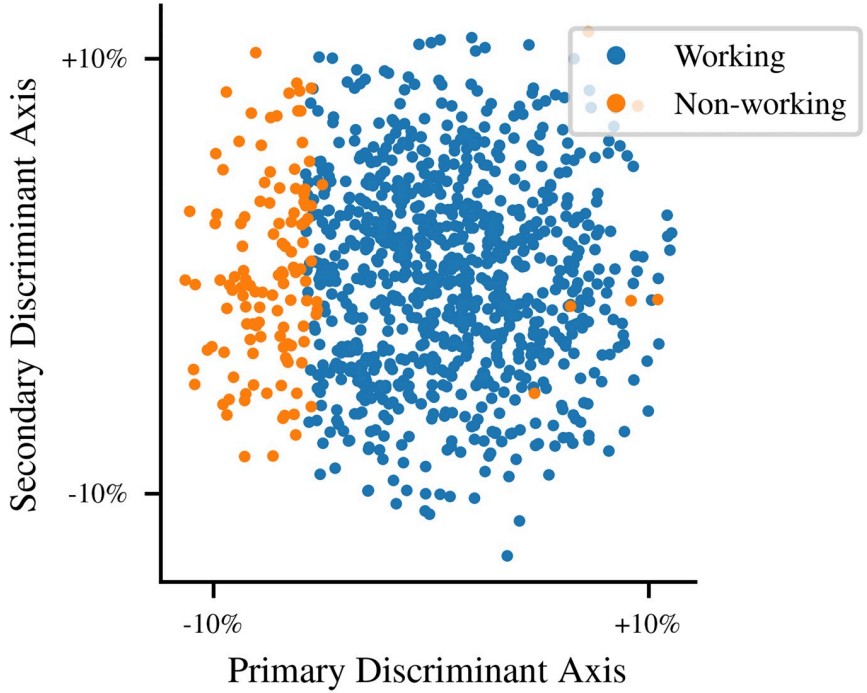

**Fig 4. Parameter variation point cloud.** A point cloud of the two-dimensional projection by LDA (described in the text) of the twelve-dimensional variable $\xi$ representing fractional variation in parameter values (increased to 15% total for illustrative purposes) for 1000 realizations of the CPG module. Blue points represent parameter values under which the module continues to function, and orange points represent those for which a failure was detected. The orange points in the majority-blue region are a result of the projection to lower dimension, rather than of inconsistent or nondeterministic behavior.

Indeed, the point cloud is nearly linearly separable, i.e. nearly every module failure can be attributed to the component of the deviation which occurred along a single axis. In this direction, a 5% deviation was sufficient to cause a failure, whereas even 15% deviation along other axes did not have this effect. The projection of the higher-dimensional point cloud onto a two-dimensional plane is shown in Fig 4. The horizontal axis of the projection is the discriminant axis returned by LDA, but the vertical axis is arbitrary because all remaining directions are equally (un)informative.

This linear discriminant axis is aligned with changes in the threshold, resting, and reversal potential parameters $V_t$, $V_r$, and $V_n$. Sensitivity to these parameters is not necessarily significant to the system design, however, because voltages in biological systems are tightly controlled through homeostatic regulation of ionic concentrations [25]. Parameters such as conductance which are not directly regulated are typically expected to vary much more with temperature [32]; these parameters have much less effect on our system behavior.

Although the points in this projection are almost linearly separable along this axis, a few points within the larger cloud correspond to a module failure but appear to be outliers because the projection has moved them away from the border of the allowed region of parameter space. In our system, just as has been observed in computational models of a lobster digestive CPG [33], the allowed region of parameter space has nontrivial higher-dimensional geometry. In particular, the same behaviors could have been achieved using entirely different parameter values corresponding to other cell types. We observe near-perfect linear separability because our initial parameter values are relatively close to a particular failure mode, but as the size of

the point cloud increases, these lower-dimensional projections become progressively less informative as they become dominated by apparent outlier points, which make up only a small fraction of the points in Fig 4.

## 4 Robotic case study

The basic motion primitives underlying locomotion are the periodic oscillations which drive different motor components. For example, the *C. elegans* locomotor system is based on a central pattern generator which produces a repeating ripple pattern down the body of the worm, propelling it forward [34]. Likewise, in many insects, each leg is associated with a CPG that produces a single step and can be modulated by higher-level control [35] as well as sensory feedback [36] to generate the full walking gait of the organism.

We can design a central pattern generation network of this type for a robotic application by combining the abstraction of asynchronous digital logic with analog effects inspired by these biological prototypes. The remainder of this paper will be focused on the design and validation of a central pattern generation network for a flexible modular robot. This robot produces a forward crawling gait, which can be viewed as a model of worm locomotion, using four linear actuators driven by a simple four-phase finite state machine.

### 4.1 The robot and its gait

The ideal robotic application for a minimal CPG is one whose locomotion is produced by rhythmic activity, but which does not need high-bandwidth control logic to stay upright. Past researchers have fulfilled this requirement with a variety of different statically stable robotic systems, in particular many different types of modular robots [37–40]. Likewise, the robotic application for our CPG is a modular robot constructed from volumetric pixels [41]. These "voxels" take the form of cubic octahedra, which can be interlocked using nuts and bolts to create a deformable lattice that in larger structures can function as a metamaterial [42]. Ten voxels are arranged into two layers in the configuration shown in Fig 5, with 3D-printed feet attached to four of the voxels on the bottom layer.

The robot electronics comprise four DC linear actuators and a control module containing custom motor drive electronics as well as a BeagleBone Black single-board computer. We use DC linear actuators rather than the linear servos which are common in similar applications because servos do not provide pose information to the software. Each of the four actuators is controlled through an L298 DC motor driver, which receives a motor direction signal and PWM enable input from the microcontroller. Each actuator also contains a linear potentiometer used as a voltage divider to produce a voltage proportional to the actuator position; the ADC reads a 10-bit fixed-point number equal to the actuator extension as a fraction of its stroke. The CPG network can use this information as a form of proprioceptive feedback in order to implement closed-loop control rather than relying on the position control built into a servo.

Our robot's walking gait is generated by four distinct, symmetrical actuation phases, each of which requires the robot to extend and contract two actuators which mirror each other while the other two are held in place. First, actuator #1 extends while actuator #3 contracts. This lifts the left front foot off the ground, while the right front foot remains planted. Second, actuator #2 extends while actuator #4 contracts; this deforms the robot's body so that the left front foot, which is no longer in contact with the ground, moves forward. During this motion, the left front foot remains in place on the ground due to the friction between the surface and the spiky 3D-printed foot, while the smooth rear feet are free to slide. Next, actuator #1

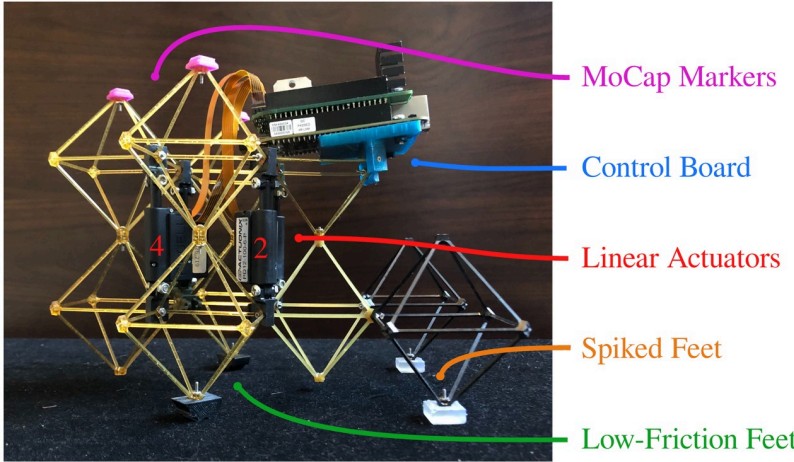

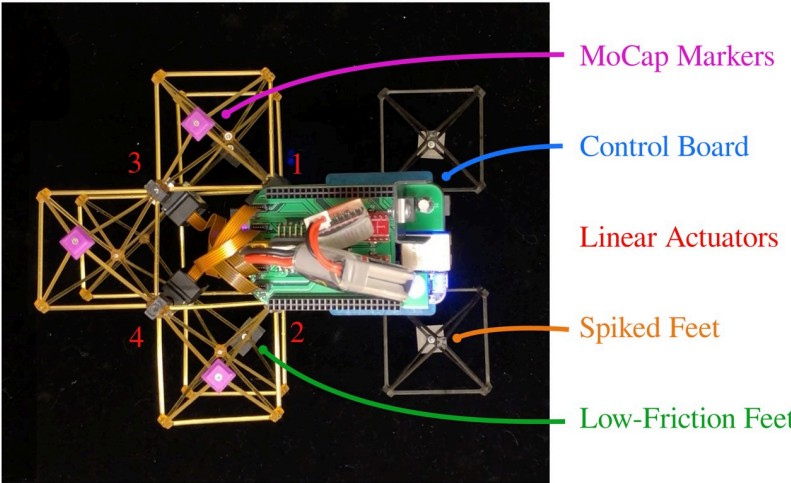

**Fig 5. The physical robot.** A diagram of the flexible modular robot for which our spiking central pattern generator is designed, shown in both side-on (above) and top-down (below) views. The three different voxel colors correspond to the three different materials described in the text.

contracts while actuator #3 extends, lowering the left front foot and raising the right. Finally, the right foot is moved forwards by contracting actuator #2 while extending actuator #4.

An advantage of the modular physical design through voxels is that individual structural subunits can be swapped out for others with different material properties—for example, our robot uses three different voxel materials. The main body is constructed from the most compliant resin so that it does not present significant resistance to the actuators, while the front feet are made from relatively rigid carbon fiber for more efficient use of the deformation generated in the body. Additionally, the node on the bottom to which the feet attach is made from a slightly stiffer resin in order to transmit motion to the feet more effectively.

## 4.2 Development of the CPG

We directly implement a state machine which controls the actuation of the original robot; four neural latch modules connected in a ring represent a one-hot encoding of the current state. Once any of the four modules is activated, for example by an external input, it gradually

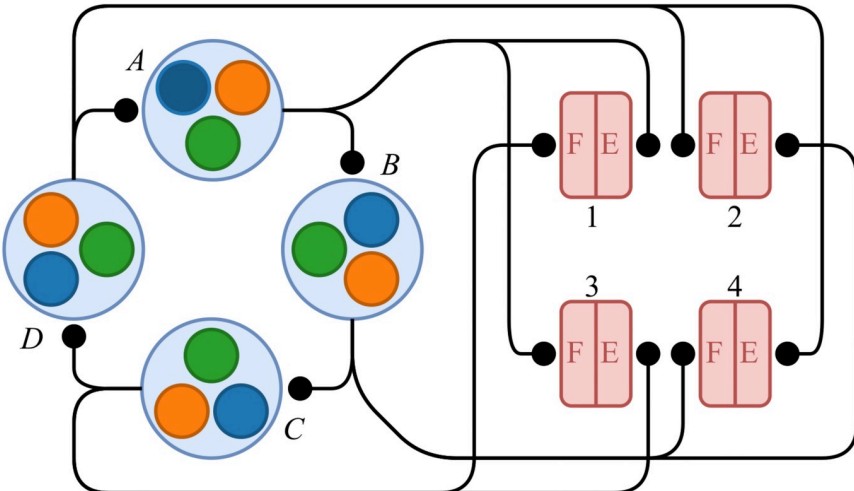

**Fig 6. The full CPG network.** The system consists of four individual modules A–D connected in a ring topology. Each module is associated with one flexor (F) and one extensor (E) among the 8 muscle cells which directly control the DC actuators. The pairs of muscle cells are labeled 1–4 in correspondence with the physical actuators. Note that the depicted connections between modules are actually bidirectional—a given module activates its successor while activating the inhibitory interneuron of its predecessor.

activates its successor, which in turn deactivates the first module. The result is a new form of oscillation; rather than the simple limit cycle of alternating firings produced by the original module, each module undergoes regular alternation between active and inactive states. Although the dynamical mechanism is not strictly bursting, similar bursts of spikes surrounded by quiescence are produced.

If we label the four modules A–D and identify them with the four states of our state machine, the system simply transitions through the states in alphabetical order. This corresponds to the four actuation phases described previously, where at any given time during the operation of the robot, one of the four actuators is expanding while another is contracting. At the same time, the other two actuators remain fixed in place in the absence of new motor input due to the mechanical properties of the worm gear rack-and-pinion actuation. As a result, with a state machine that cycles through four states, the desired pattern of muscle actuation is generated simply by having each state excite the corresponding extensor and the opposite flexor. The result is the full network depicted in Fig 6.

This network implements the correct state machine, but the speed with which it switches between states is parameter-dependent and much too fast to produce the desired gait. The inertia and damping inherent to the mechanical construction of the robot mean that the actuators simply oscillate in place without producing any useful forward motion. For this reason, it becomes necessary to use feedback to modulate the speed with which the network switches between states. One can imagine that, as in *C. elegans* [34], the neurons of our CPG are equipped with strain-sensitive neurites which create an inhibitory leakage current in proportion to the deviation of the preceding module's actuator from its target position. This prevents the initiation of the next state when the current state has not yet achieved its intended actuator position.

In order to describe the proprioceptive feedback $I_{\mathrm{prop},i}$ to the $i$th module due to this linear feedback current, we introduce the actuator position variables $z_i$ and $z_{i^*}$ representing the relative extension of the previous actuator and its opposite counterpart as numbers between 0 and 1. Additionally, there is a feedback constant $k_p$ which must be tuned by a bit of trial and error

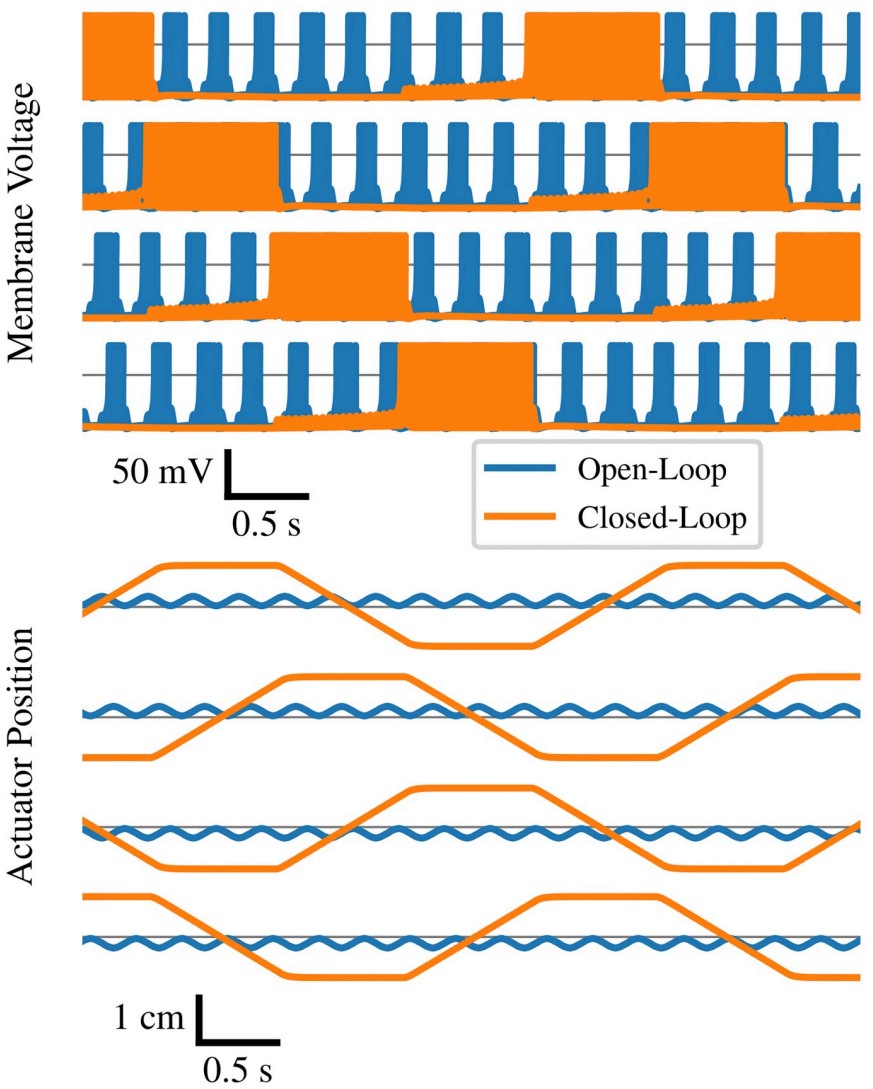

**Fig 7. Open vs. closed loop.** Comparison in simulation between the open-loop and closed-loop oscillatory behavior of the designed neural network. The top four traces represent overall neural activity in each of the four modules during the two experimental conditions, and the lower four traces represent the extension of the actuators.

but which depends only on the neuron parameters, not the physical properties of the robot. In our application, a value $k_p = 25$ pA was effective. The resulting feedback current is given by:

$$\begin{aligned} I_{\text{prop},i} &= -k_p|1 - z_i| - k_p|z_{i*}| \\ &= -k_p(1 + z_{i*} - z_i) \end{aligned} \tag{3}$$

The behavior of the open-loop and closed-loop networks are contrasted in Fig 7. Both network configurations produce the same pattern of alternation between states as well as the same pattern of actuation, but the short duration for which each state remains active in the open-loop network means that the actuators only extend a very short distance before being forced to contract again. Note that this demonstration was performed in simulation for clarity; the depicted actuator position is derived from treating the actuator kinetics as ideal linear damped masses. In the realistic case, the difference is more extreme because frictional losses cause the

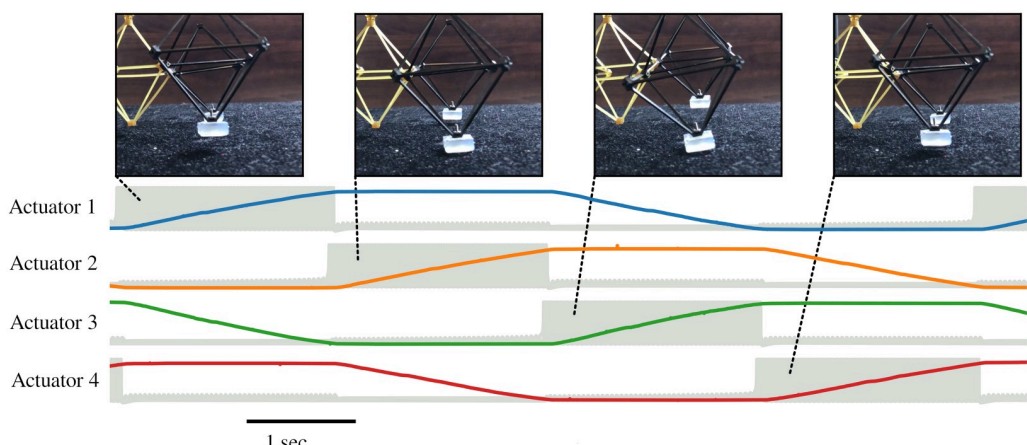

**Fig 8. Robot gait time-lapse.** A time-lapse of the motion of the robot's front feet, overlaid on top of an abstract representation of the neural activity necessary to produce the depicted motion. The gray traces are overlays of the membrane voltage of all three neurons in each module, which disappear into each other due to the long timescale, and the four colored traces represent the extension of the four actuators. The four photographic insets correspond to the four phases of actuation described in the text. First, the right foot is raised above the ground as it moves forward. Next, the right foot comes down and both feet are briefly planted. Then, the left foot is raised and begins to move forward. Finally, the left foot is lowered to the ground again, and the cycle continues.

open-loop excursion of the actuators to be significantly less than pictured here. As a result, the robot under open-loop control moves extremely inefficiently or not at all.

## 5 Physical experiment

To this point, all description of the full spiking neural network depicted in Fig 6 has been simulated, qualitative, or both. In order to provide a concrete physical demonstration of the efficacy of our approach, in this section we describe the experiments which were carried out in order to validate the physical robotic platform just described.

During all of these physical experiments, we recorded the full state of the neural network as well as the commanded actuator effort and outputs of the proprioceptive sensors. This enables us to directly correlate the activity of the neural network with the physical behavior of the robot, as in Fig 8.

In order to quantify the physical movement of our robot, we implemented a simple motion capture system through color-based particle tracking, with three magenta 3D-printed markers affixed to the top of the robot. We assume that once the camera is calibrated to minimize distortion, position in the image plane is proportional to physical position in the ground plane. This assumption allows us to roughly calibrate the system using the fact that the distance between the motion capture markers while the robot is in a neutral pose is determined by the known geometry of the voxels.

An example of the type of data which can be captured by this system is presented in Fig 9, where the motion of the markers during forward locomotion is plotted on top of an image of the robot's position at the end of the recording. The physical position of the robot during this behavior is shown in Fig 10; with this motion-tracking data, we can approximate the robot's forward speed to be 3 cm/min, mainly limited by the slew rate of the actuators.

Next, we implemented a variation on our controller which is capable of reversing direction in response to an aversive stimulus. In a simple network like this, the easiest way to implement two different locomotive directions is the route taken by *C. elegans*: introducing what amounts

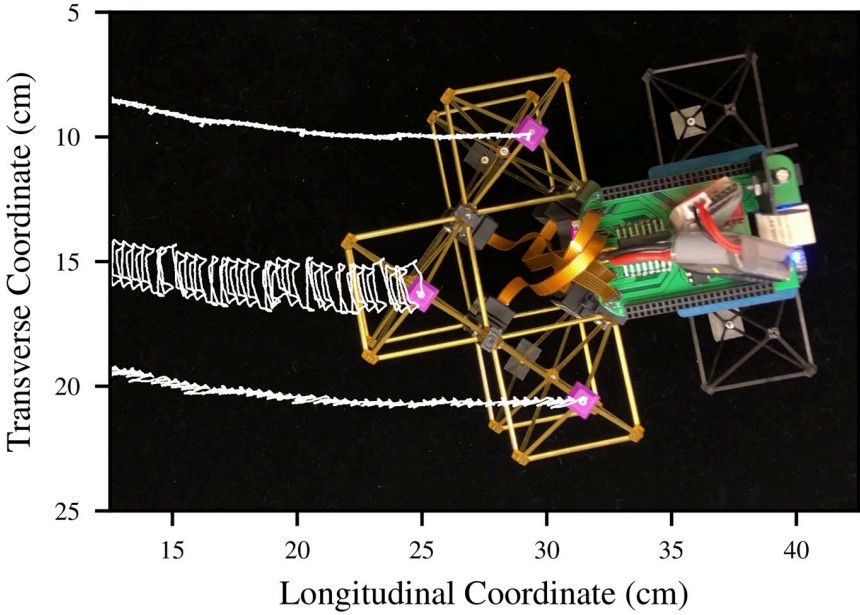

**Fig 9. Overhead motion capture.** Motion capture traces demonstrating the movement in the horizontal plane (from right to left) of three plastic motion capture markers affixed to the top of the robot. White traces represent the position of the purple markers over approximately five minutes. The oscillatory patterns in the white traces are not noise in the motion capture system, but rather represent the cyclic trajectory traced out by the voxel nodes.

to two parallel copies of the same CPG network [34], one of which is active during forward movement and one of which is active when the robot is moving backward. We then introduce a single neuron responsible for detecting the presence of some aversive stimulus and signaling that the forward network should be deactivated and the reverse network activated.

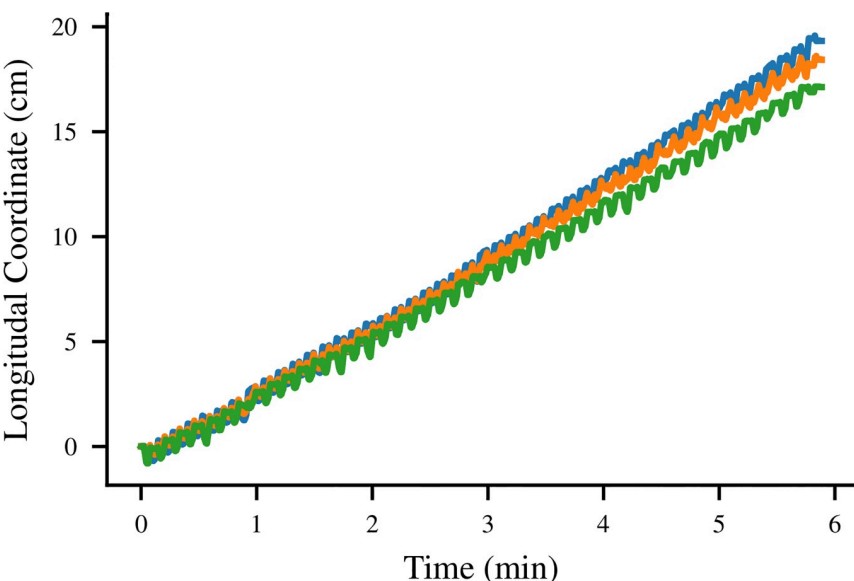

**Fig 10. Forward motion of the robot.** The forward displacement of the three motion capture markers from their starting positions during the motion displayed schematically in Fig 9.

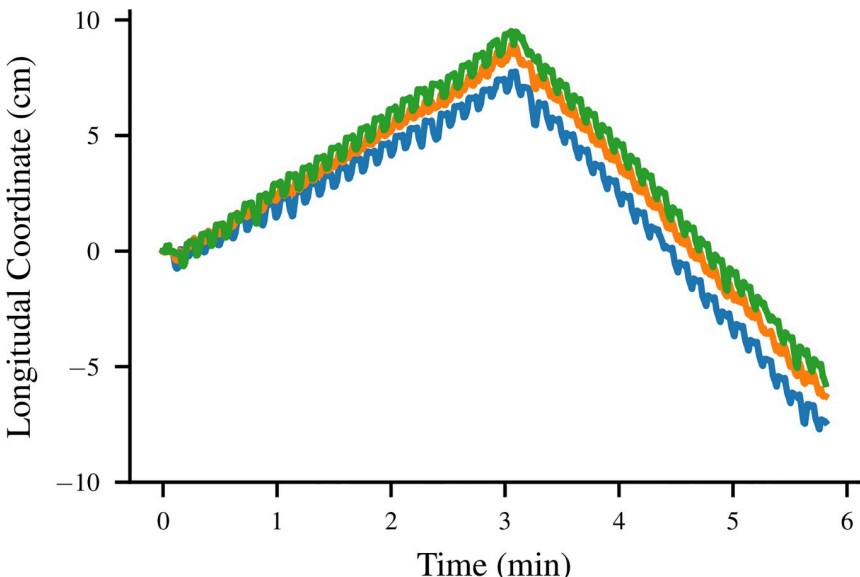

**Fig 11. Reflexive reversal of the robot.** The forward displacement of the three motion capture markers from their starting positions during the reversal reflex experiment.

The result is that as soon as an aversive stimulus is detected during forward motion, the robot reverses direction and begins to move backward instead. In our experiments, the "aversive stimulus" of choice occurred at an arbitrary time, but it would be simple to replace this behavior with something more reactive, such as a physical bumper to detect obstacles. The position of the robot over time during this experiment is shown in Fig 11.

# 6 Conclusion

We have described a simple approach to the design of spiking neural networks for robotic control. By taking advantage of the fact that the behavior of neurons can be approximated by a simple binary model we can design a connectome capable of producing the desired behavior. A case study was provided by a flexible modular robot, where we applied this approach to produce a tripod-stable gait which can be modulated by sensory feedback or external inputs much like a biological central pattern generator.

In the future, we are interested in developing a method to efficiently characterize the geometry of the extended region of parameter space in which the designed neural module continues to function. This would allow a more thorough approach to the question of robustness vs. modulation, as the acceptable region of parameter space is clearly significantly larger and more irregularly shaped than the region studied here. It would also be of interest to more rigorously study the question of how single-module robustness translates to the full system.

Additionally, in the real-world evolution of central pattern generation networks, although the simple underlying architecture tends to be evolutionarily conserved, evolution tends to provide a great deal of variety in the ways that this architecture can be modulated by higher-level control [43]. We seek to explore such additional forms of modulation in future work; for example, we may introduce a higher-level learned or evolved network which interacts with the world by modulating or controlling an underlying conserved central pattern generation network.

## Supporting information

**S1 Code. Simulation code.** A Jupyter notebook containing the Julia code which implements our simulations as well as the mathematical results of Section 3.
(IPYNB)

**S1 Video. Forward locomotion.** A video of the robot performing forward locomotion, from which the motion capture data in Figs 9 and 10 were computed.
(MOV)

**S2 Video. Backward locomotion.** A video of the robot performing backward locomotion, from which the position data depicted in Fig 11 were computed.
(MOV)

**S3 Video. Side view.** Side view of the robot taking a few steps, used to generate Fig 8.
(MOV)

## Acknowledgments

The authors would like to acknowledge the technical support of the Braingeneers research group as well as a donation made possible by Eric and Wendy Schmidt by recommendation of the Schmidt Futures program.

## Author Contributions

**Conceptualization:** Alex Spaeth, Maryam Tebyani, David Haussler, Mircea Teodorescu.

**Data curation:** Alex Spaeth, Maryam Tebyani.

**Formal analysis:** Alex Spaeth.

**Funding acquisition:** David Haussler, Mircea Teodorescu.

**Investigation:** Alex Spaeth, Maryam Tebyani.

**Methodology:** Alex Spaeth, Maryam Tebyani.

**Software:** Alex Spaeth, Maryam Tebyani.

**Supervision:** David Haussler, Mircea Teodorescu.

**Validation:** Maryam Tebyani.

**Visualization:** Alex Spaeth.

**Writing – original draft:** Alex Spaeth.

**Writing – review & editing:** Alex Spaeth, Maryam Tebyani, David Haussler, Mircea Teodorescu.

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
