## [Decision Letter · Decision Letter 0]

2 Sep 2020

PONE-D-20-21490

Spiking neural state machine for gait frequency entrainment in a flexible modular robot

PLOS ONE

Dear Dr. Spaeth,

Thank you for submitting your manuscript to PLOS ONE. After careful consideration, we feel that it has merit but does not fully meet PLOS ONE’s publication criteria as it currently stands. Therefore, we invite you to submit a revised version of the manuscript that addresses the points raised during the review process.

The manuscript requires revision to provide details of the model and its justification, and should be put into the context of the contemporary literature.

 The introduction makes it seem that computation with spiking neural networks is a novel concept (by contrasting it to ANNs in the second paragraph of the introduction but not mentioning prior literature) developed here. - This needs major rephrasing.

 The mechanisms of rhythmogenesis in the presented controller is seemingly based on temporal summation of synaptic inputs and the individual modular building blocks are not intrinsically rhythmogenic. While this is a possible mechanism for the generation of rhythmicity, there are many others and it is not the most likely one. In mammals, for example, the view is that each limb-specific circuit is capable to produce a rhythm itself (since a rhythm can be elicited in the hemicord) and slow persistent ion channels likely underly rhythm genesis. The chosen mechanism has to be discussed in connection with the literature.

The article refers to the individual modules as oscillator modules or CPG modules and even CPG networks, which is clearly misleading. The two types of oscillations described in the paper, tonic neural firing vs. neural bursting, should be clearly distinguished in the text.

 Figures showing neural activities should include the synaptic currents to illustrate the mechanisms of rhythmogensis (Figs 2, 7).

 It is not clear why the robustness and sensitivity of only the three-cell building block is analyzed.

How function of the latch-circuit was operationalized for the sensitivity/discriminant analysis? Was only the presence of a limit-cycle (> 5ms) used= What about the ability to change states using external inputs?

 Results are mostly described qualitatively and lack quantitative detail (e.g., what are the axes in Fig 4).

The results should be presented in detail so that they would be reproducible. Parameters used in Fig 3 are not given etc. All parameters should be clearly specified and easy to relate to the described results. Furthermore, the source code for all simulations (including all figures) should be published with the paper (e.g. GitHub, ModelDB).

Introduction: Neuromorphic computation is introduced as using spikes. In the next paragraph neuromorphic machine learning is equated to artificial neural networks. This is confusing.

Line 123: “model parameters have biophysical meaning” - This is a bit of a stretch, esp. when compared to Hodgkin-Huxley type models.

Lines 169-172: The selected model parameterizations taken from Izhikevich should be explained in more detail. What is their spiking patterns. Why were they selected.

Line 203ff: This is an odd analogy. Especially since that would relate to the original non-leaky integrate and fire neuron. While the neuron at hand is the extension of the leaky version.

Fig 3: It is not clear what the authors try to illustrate. At the very least the exact parameters used should be indicated.

We look forward to receiving your revised manuscript.

Kind regards,

Gennady Cymbalyuk, Ph.D.

Academic Editor

PLOS ONE

Journal Requirements:

"This research was supported by a grant made to the Braingeneers research group by

Schmidt Family Futures.

The funders had no role in study design, data collection and analysis, decision to

publish, or preparation of the manuscript."

We note that you received funding from a commercial source: "Schmidt Family Futures"

Reviewers' comments:

Reviewer's Responses to Questions

**Comments to the Author**

1. Is the manuscript technically sound, and do the data support the conclusions?

Reviewer #1: Yes

Reviewer #2: Partly

2. Has the statistical analysis been performed appropriately and rigorously? 

Reviewer #1: N/A

Reviewer #2: No

3. Have the authors made all data underlying the findings in their manuscript fully available?

Reviewer #1: Yes

Reviewer #2: No

4. Is the manuscript presented in an intelligible fashion and written in standard English?

Reviewer #1: Yes

Reviewer #2: Yes

5. Review Comments to the Author

Reviewer #1: This is a very interesting paper reporting ‘modular architecture for neuromorphic closed-loop control based on bistable relaxation oscillator modules’. Authors have designed ‘neural state machine’ controller that was used to control legged robot locomotion. This controller consists of 4 modules organized in circle. Each module represents 3 neuron unit burst generator, two neurons are reciprocally activating each other thus producing spike oscillations while third neuron regulating start and stop of spiking trains by inhibiting both these neurons. A robot constructed by authors has 4 legs controlled by 4 linear actuators. Each linear actuator is controlled by two opposite CPG modules in the ring providing command to flex or extend robot leg.

This is a very well performed implication of Unit-Burst-Generator hypothesis for CPG organization by Grillner to a robot and meticulously analyzed sensitivity of the model parameters shows robustness of proposed controller.

The paper describes that robot models legged locomotion, that is much more complex behavior than just flexion/extension of legs. It looks like model of worm locomotion for me.

The paper would be stronger if the authors could avoid using electrical engineer jargon. Examples: line 17 ‘converted ANNs’; line 52 ‘SR latch’.

Minor

Lines 30-31: For the sake of completeness, the discussion of the bistability in neuron models should include also the study by Dashevskiy T, Cymbalyuk G.Front Comput Neurosci. 2018, Barnett W, O'Brien G, Cymbalyuk G.J Neurosci Methods. 2013, Malashchenko T, Shilnikov A, Cymbalyuk G.PLoS One. 2011;6(7), Shilnikov A, Calabrese RL, Cymbalyuk G.Phys Rev E Stat Nonlin Soft Matter Phys. 2005

Reviewer #2: Spaeth et al. outline an approach to construct a state-machine using modules of spiking neural circuits. They describe bifurcations and analyze the robustness of SR-latch-like circuits. Then they assemble them into a state-machine and use it to control a four-legged crawling robot. They argue that these circuits resemble those of a CPG in the biological system.

The paper is well written and generally well thought out but also has some major issues: it is not properly put in the context of prior literature and lacks crucial detail in the description of the results. My detailed comments are listed in the following:

1 There is a large body of literature describing models of locomotor CPGs using populations of spiking neurons, such as the work of Eve Marder and Ilya Rybak and others. These warrant discussion. As the paper currently reads, a naive reader could think that the use of spiking neurons for CPG models is a novelty of the paper.

1a Further, there are several examples of spiking neural networks used for the control of robots (a simple google scholar search will reveal several).

1b The biggest issue of the paper is, that the introduction makes it seem that computation with spiking neural networks is a novel concept (by contrasting it to ANNs in the second paragraph of the introduction but not mentioning prior literature) developed here. - This needs major rephrasing.

2 The mechanisms of rhythmogenesis in the presented controller is seemingly based on temporal summation of synaptic inputs and the individual modular building blocks are not intrinsically rhythmogenic. While this is a possible mechanism for the generation of rhythmicity, there are many others and it is not the most likely one. In mammals, for example, the view is that each limb-specific circuit is capable to produce a rhythm itself (since a rhythm can be elicited in the hemicord) and slow persistent ion channels likely underly rhythm genesis. The chosen mechanism has to be discussed in connection with the literature.

2a The article refers to the individual modules as oscillator modules or CPG modules and even CPG networks, which is clearly misleading. The two types of oscillations described in the paper, tonic neural firing vs. neural bursting, should be clearly distinguished in the text.

2b Figures showing neural activities should include the synaptic currents to illustrate the mechanisms of rhythmogensis (Figs 2, 7).

3 It is not clear why the robustness and sensitivity of only the three-cell building block is analyzed. The robustness of the final network would be of much greater interest, especially since the single module seems to lack the ability to generate bursts.

3a It is not exactly clear how function of the latch-circuit was operationalized for the sensitivity/discriminant analysis. Was only the presence of a limit-cycle (> 5ms) used= What about the ability to change states using external inputs?

3b Results are mostly described qualitatively and lack quantitative detail (e.g., what are the axes in Fig 4).

4 The results are unlikely to be reproducible. For example, I couldn’t find the parameters for the synaptic conductances between the modules, parameters used in Fig 3 are not given etc. All parameters should be clearly specified and easy to relate to the described results. Furthermore, the source code for all simulations (including all figures) should be published with the paper (e.g. GitHub, ModelDB).

Some minor comments:

Introduction: Neuromorphic computation is introduced as using spikes. In the next paragraph neuromorphic machine learning is equated to artificial neural networks. This is confusing.

Line 123: “model parameters have biophysical meaning” - This is a bit of a stretch, esp. when compared to Hodgkin-Huxley type models.

Lines 169-172: The selected model parameterizations taken from Izhikevich should be explained in more detail. What is their spiking patterns. Why were they selected.

Line 203ff: This is an odd analogy. Especially since that would relate to the original non-leaky integrate and fire neuron. While the neuron at hand is the extension of the leaky version.

Fig 3: It is not clear to me what the authors try to illustrate. At the very least the exact parameters used should be indicated.

In summary, the paper has some value but needs major revision. Mainly the objectives of the paper have to be discussed in the context of current literature (spiking neural networks are not a novelty, neither for engineering purposes nor for models of CPGs; the novelty is the modular latch-like structure) and the rigor in the presentation of the model and analysis results has to be improved.

6. PLOS authors have the option to publish the peer review history of their article (what does this mean?). If published, this will include your full peer review and any attached files.

Reviewer #1: No

Reviewer #2: No

---

## [Author Response · Author response to Decision Letter 0]

18 Sep 2020

The text in general has been rewritten and substantially clarified, in order to better justify its position in the context of the literature as well as to clarify many of the excellent points made by the reviewers. Additionally, a new SI file has been included giving all of the simulation code in Jupyter notebook format.

The specific changes which have been made are addressed in more detail in the uploaded review response table, `responses.pdf` of our resubmission.

---

## [Decision Letter · Decision Letter 1]

23 Sep 2020

Spiking neural state machine for gait frequency entrainment in a flexible modular robot

PONE-D-20-21490R1

Dear Dr. Spaeth,

We’re pleased to inform you that your manuscript has been judged scientifically suitable for publication and will be formally accepted for publication once it meets all outstanding technical requirements.

Kind regards,

Gennady Cymbalyuk, Ph.D.

Academic Editor

PLOS ONE

Additional Editor Comments (optional):

Reviewers' comments:

Reviewer's Responses to Questions

**Comments to the Author**

1. If the authors have adequately addressed your comments raised in a previous round of review and you feel that this manuscript is now acceptable for publication, you may indicate that here to bypass the “Comments to the Author” section, enter your conflict of interest statement in the “Confidential to Editor” section, and submit your "Accept" recommendation.

Reviewer #2: All comments have been addressed

2. Is the manuscript technically sound, and do the data support the conclusions?

Reviewer #2: Yes

3. Has the statistical analysis been performed appropriately and rigorously? 

Reviewer #2: Yes

4. Have the authors made all data underlying the findings in their manuscript fully available?

Reviewer #2: Yes

5. Is the manuscript presented in an intelligible fashion and written in standard English?

Reviewer #2: Yes

6. Review Comments to the Author

Reviewer #2: The revised version is a clear improvement of the initial submission. The authors addressed all concerns raised by me and the other reviewer. I would like to congratulate the authors for their work and recommend this revised version of the article for publication.

7. PLOS authors have the option to publish the peer review history of their article (what does this mean?). If published, this will include your full peer review and any attached files.

Reviewer #2: **Yes: **Simon M. Danner

---

## [Editor Report · Acceptance letter]

25 Sep 2020

PONE-D-20-21490R1 

Spiking neural state machine for gait frequency entrainment in a flexible modular robot 

Dear Dr. Spaeth:

I'm pleased to inform you that your manuscript has been deemed suitable for publication in PLOS ONE. Congratulations! Your manuscript is now with our production department. 

Kind regards, 

on behalf of

Dr. Gennady Cymbalyuk 

Academic Editor

PLOS ONE